# Strategies to enhance the level of service and safety of rural roads: A case study

Qiannan Ai[1], Jun Zhang[2]*, Yuling Ye[3,4]

1 School of Public Utility, Jiangsu Urban and Rural Construction College, Changzhou, Jiangsu Province, China, 2 Department of Transportation Engineering, College of Architectural Science and Engineering, Yangzhou University, Yangzhou, Jiangsu Province, China, 3 College of Transportation Engineering, Tongji University, Shanghai, China, 4 Key Laboratory of Road and Traffic Engineering of the State Ministry of Education, Shanghai, China

* Zhangjun93@yzu.edu.cn

## Abstract

Faced with the contradiction between the increasing traffic volume and the aging road infrastructures in the rural area, this paper aims to propose feasible strategies to enhance the level of service and safety, by a case study of the rural area in the north Jintan district. In order to figure out current issues related to rural roads, a carefully designed investigation has been conducted, and the results of the two-week investigation include roads' basic information, traffic signs and protective facilities, surrounding landscape, and etc. Based on the field driving tests, specific problems including signs category, signs installation and facility maintenance have been fully analyzed. Meanwhile, the problem of roadnet connectivity has also been pointed out through the theory of complex network, and results show that the average node clustering coefficient and shortest path length perform worse than the demonstration plot of other rural districts. For the sake of rural traffic safety and management efficiency, both quantified and qualified strategies have been put forward. The quantified strategies include the regular inspection indicators, the safety sight distance at T-type crossings, as well as the risk severity of sections and the crossings. The qualified strategies involve the management of trucks and roadworks, the setting of signalized intersections, and the timely updates of traffic signs and facilities. Finally, an intelligent management system framework has been established for rural road traffic, with highly interconnected modules of data acquisition, risk identification and information publishing.

## Introduction

The increasing rural economy under the integration of urban and rural development has brought the increasing traffic demand on rural roads, which has become a challenge to the transportation safety and rural governance. In most rural areas of China, many factors contribute to the urgency of enhancing rural road traffic, including the rural roads construction oversight, the limitation of infrastructure maintenance, and increasing traffic conflicts or crashes between local residents and passing-by vehicles. Under the national strategy of all-around

**Data Availability Statement:** The major data supporting the findings has already shown in the manuscript, and the other data including the GIS files of regional road network and the traffic characteristics data are available on request from the Chanzhou City Planning and Design Institute,

email: jsczghy@126.com, the data are not publicly available due to privacy restrictions.

**Funding:** Local Innovation Talent Project of Yangzhou under Grant number 2022YXBS118. Funder: Yangzhou Government, Jiangsu Province, China. Recipient: Jun Zhang The funders had no role in study design, data collection and analysis, decision to publish, or preparation of the manuscript.

**Competing interests:** The authors have declared that no competing interests exist.

rural revitalization, the improvement of rural transportation functions as a key support, wherein establishing a lifecycle mechanism of coordinated construction, management, maintenance and operation has been put on the agenda of regional governors. The first step is to screen current rural road system for potential risks and deficiencies, and put forward possible enhancement strategies for daily managing and future planning. Generally, current literatures on the rural roads enhancement can be divided into three categories. The first is the basic infrastructure improvement, the second is the traffic safety facilities configuration, and the third is the advanced management strategies.

Studies on the improvement of basic infrastructures mainly include the road alignment and the road pavement. Road alignment improvement focuses on the interactions between road geometric parameters and driving behaviors. Based on the data of geometric alignment and surface performance of a two-lane rural road, Bella explored the corresponding influence on driving safety via theories of decision tree and Bayesian network [1]. Machiani et al. further studied drivers' perception of curves upon an analysis of speed and lateral position on the rural roads [2]. It has also been revealed that lane width, curve length and curve radius are key factors affecting the traffic order on the rural road, using the safety performance functions [3]. By comparison, the road pavement improvement considers more on a sustainable design and maintenance. To realize the environment-friendly maintenance, Kumar and Ryntathiang studied the performance of the pavement condition index with the age of pavement, and introduced the technology of micro-surfacing [4]. Pasindu et al. performed an optimization analysis for the pavement management of the rural roadnet, upon the calculation of cumulative safety and international roughness [5]. Current literatures on the enhancement of basic infrastructure for rural roads are safety-based geometric design and pavement management, with common considerations on traffic volume, traffic composition and design speed [6, 7], and the results can provide important references for enhancement strategies.

As to the traffic safety facilities configuration, the signs, markings, signals, and protective infrastructures are usually discussed. Griselda et al. pointed out that the completeness and correctness of markings or signs plays an important role in reducing crashes on rural highway [8]. Zhang et al. studied the visual recognizability of different traffic signs, and analyzed the impact of sight distance, lane location, occluded ratio, shape damage, and installation parameters (height, angle, size, and etc.), which can provide a solid base for regulations modification [9]. Reinolsmann et al. tested the warning distance of variable message signs on rural roads with low visibility, and designed a visualized sandstorm animation to warn drivers [10]. The method of setting traffic safety facilities for rural highway in different road sections (small curve, steep slope, rural tunnel, crossings and etc.) have also been proposed, including traffic signs, traffic markings and anti-collision facilities [11]. It can be concluded that the road alignment and traffic environment are a key basis for the configuration of safety facilities on rural roads. The configuration schemes can be improved to the level above standards by analyzing the interaction mechanisms among facility installation parameters, drivers' perception attributes and driving behaviors.

In the aspect of advanced management strategies, a variety of updated technologies have been proposed or preliminarily applied. The safety evaluation and enhancement for road infrastructures are critical parts during daily management [12, 13]. Vaiana et al. carried out a safety performance evaluation through the RSI approach for a high-risk rural road [14]. Faced with the deterioration of rural road infrastructure, Subedi et al. developed the priority list of road maintenance considering economic factors and technological factors [15]. Similarly, Gupta et al. further assessed the maintenance status by structural and functional parameters including pavement roughness, cracking, texture depth [16]. Meanwhile, the intelligent traffic management in the rural area has gained more attention recently. Targeting at providing effective

information for travelers, Rasaizadi et al. proposed an ELP (ensemble learning process) based prediction method of traffic distribution on rural roads [17]. Considering the difference between driving perception and action, Tian et al. conduct an investigation on the Rural Intersection Conflict Warning System, and provided suggestions for delicate management [18]. David et al. proposed a model of measuring real-time carbon emissions for rural road, for the convenience of monitoring roadside air pollutions [19]. Generally, existing strategies have covered the lifecycle management of rural roads including planning, construction, operation and maintenance, but the strategies are independent from each other, the integration of different monitoring and measuring data sources appears to be the development trend for the smart management and decision.

From the foregoing literatures, it can be concluded that the enhancement of rural roads is a systematic work, while most existing literatures focus on presenting a specific method or strategy to improve the level of service or safety, it is undeniable that the proposed strategies are theoretically feasible and useful, but the methods of different studies are usually mutual separated to each other, causing the problem of global evaluation and enhancement when it comes a rural network. Most theories are not appropriate for use in rural areas due to the absence of financial support and professional engineers for rural roads. The current gap is in evaluating the status of rural roads effectively and adopting appropriate enhancement strategies correctly, taking into account their feasibility. This paper aims to illustrate the implementation method of field investigating and countermeasures analyzing, through a case study of rural roads in Jintan District, Jiangsu Province, China. Our paper is noteworthy for its presentation of a well-organized enhancement mechanism for rural roads, which consists of both practical suggestions and theoretical assessments.

The remainder of this paper is structured as follows. The Current situation investigation section describes the current situation of rural roads in the studied area, mainly introducing the road information and infrastructure characteristics. The section of Road infrastructure condition analysis performs a condition analysis of road infrastructures including traffic signs, traffic control, facility maintenance and network topology. An enhancement mechanism is presented in the Enhancement mechanism section, followed by an architecture framework of intelligent management system for rural roads. Finally, the Conclusions section ends the paper with major contributions and possible future work.

## Current situation investigation

### Data investigation

Considering the peculiarity of rural roads, a general framework of field investigation is designed in Fig 1, composed of three aspects, the basic road information, the traffic safety facility, and the roadside landscape. Among the three, the traffic safety facility has strong relationships with the other two, wherein the traffic signs and protective facilities are dependent on the road section attributes and the surrounding circumstances.

Taking into account the demands of future analysis and studies, the data collection methods employed in field observations and measurement are:

1. Photographing method. Taking photos of roadside environment, road pavement and traffic-related facilities. The photos are used as basic references to illustrate current condition of rural roads.

2. Aerial video recording. Recording the traffic status of rural road sections and signal-control intersections with UAVs (unmanned aerial vehicle). The aerial videos are the basis of road traffic analysis, including traffic volume, traffic density and conflicts distribution.

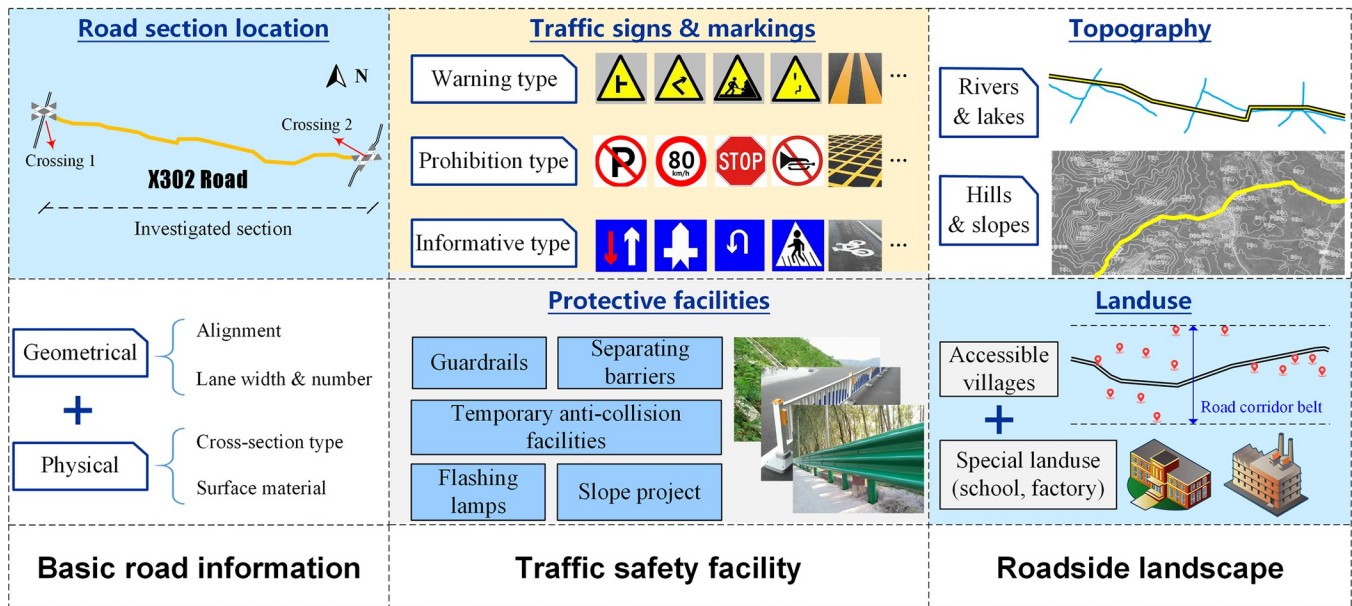

**Fig 1. The designed investigation content for rural road sections.**

3. Positioning method. Using GIS map tools to mark the locations of inspected infrastructures, facilities, and signs in synchronization with the photographing method.

4. Trajectory recording method. Recording the trajectory data during the repeated paths driving test, which is utilized for the speed distribution analysis and travel time estimation. Dynamic path management for trucks and traffic guidance for reconstruction sections can be aided by the trajectory data.

5. Physical Measurement. Using tape measures, it is possible to measure the physical parameters of road infrastructures and signs, including lane width, sign size and location, guardrail length, and other parameters. Safety checks and evaluations of traffic facilities depend heavily on measurement data.

6. History data investigation. Obtaining the data like area population distribution, road maintenance records, road accidents record from the published data of local government and transportation sectors. Indicators calculation and prediction can be utilized with these types of data, particularly for decision making in traffic control and infrastructure maintenance.

## Roads information

The area of study is the northern rural zone of Jintan district, situated in the southwest of Changzhou, China. The field investigation was conducted from 1st to 22nd, Jun 2023, approved by the Science and Technology Bureau of Changzhou, where the major roads and key connecting branches are involved, excluding the ways accessing villages, as shown in Fig 2. The major information of major investigated rural roads is indicated in Table 1. All roads in the study area share a same climate condition, the average daily rainfall is about 2.91 mm, where days with rainfall over 10 mm only accounts for 2.85% of the whole year. Meanwhile, the average wind speed is about 3.3 m/s, and the average daily sunshine percentage is about 46%.

In Table 1, the types of cross-section types are classified into four kinds according to the actual situation of the studied area, as indicated in Table 2.

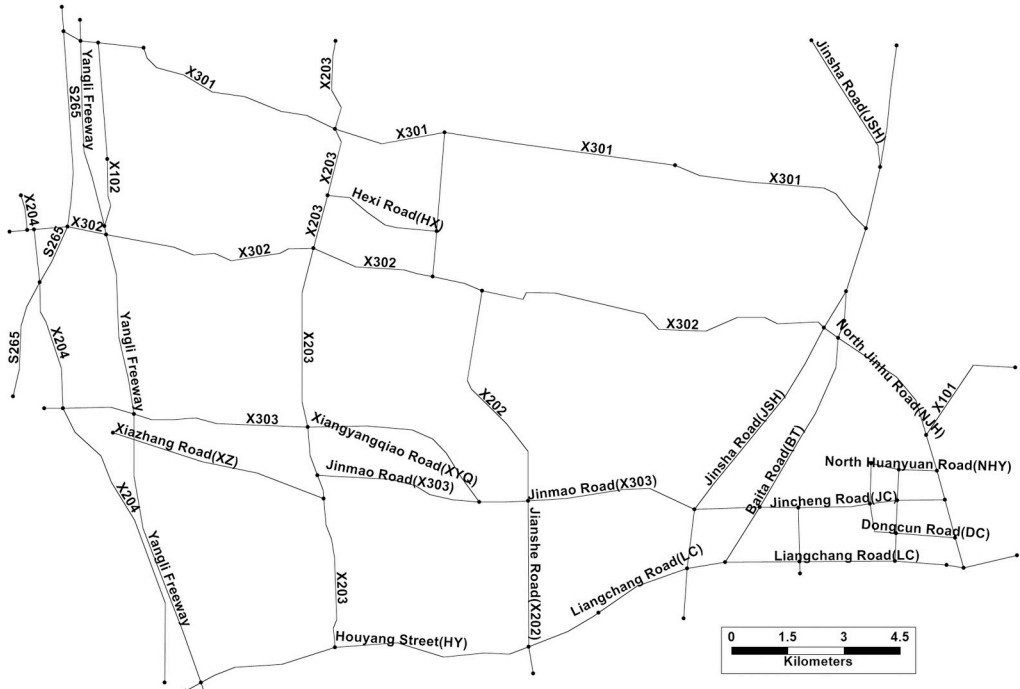

**Fig 2. Rural roads network within the study area.** (Republished from TransCAD under a CC BY license, with permission from Caliper Corporation, original copyright 2024).

The following three points summarize the general condition of rural roads within the study area based on the investigated data.

1. 29% of the studied road sections are dual carriageways designed with bi-directional 4 motor lanes, where the central isolation forms are different. E.g., the Jinsha road adopts the central green belt to isolate contrary carriageways, while the North Huanyuan road uses the guardrail, as shown in Fig 3.

2. 84% of the investigated road sections are the asphalt pavement, which can provide a better driving comfortability. As shown in Fig 4, the rest of the road sections are paved with cement, which presents serious problems with cracks and depressions.

3. 5% of the road sections are triple carriageways, distributed along the Jinzhuang Road and the X203 Road, especially the sections passing through the town area, as indicated in Fig 5. Due to the large demand of non-motorized traffic, barriers have been installed to isolate the motor lanes from the bilateral non-motorized lanes, in order to guarantee the safety of traffic participants.

### Traffic signs and safety facilities

Three representative roads, Baita Road, X302, and Jianshe Road, were chosen to reflect the traffic facility distribution on rural roads.

**Baita Road.** The investigated road section spans from the BT-JC intersection to the BT-NJH intersection, with a total length of 4.4 km. Two motor lanes and two non-motorized lanes are located on a single carriageway in the investigated section. The cross section parameters together with traffic signs and facilities are indicated in Fig 6, where the major facilities are warning signs, prohibition signs and protection infrastructures.

**Table 1. Basic information of major road sections.**

| Road | Section | Length (km) | Cross-section parameters | | Traffic | | Major land use |
|---|---|---|---|---|---|---|---|
| | | | Type | Lane distribution and surface pavement | Volume (PCU/day) | Truck ratio | |
| X301 | JSH-X203 | 12.79 | 1 | Motor: 3.5m×2 Non-motor: 1.5m×2 (Asphalt) | 1130 | 6.3% | Agriculture |
| | X203-X102 | 5.04 | 1 | | 610 | 4.7% | |
| X302 | JSH-X202 | 8.38 | 1 | Motor: 3.5m×2 Non-motor: 1.5m×2 (Asphalt) | 1560 | 8.1% | Agriculture |
| | X202-JZ | 1.19 | 1 | | 2410 | 13.6% | |
| | JZ-X203 | 2.85 | 1 | | 2050 | 15.1% | |
| | X203 west | 4.89 | 1 | | 1520 | 12.4% | |
| X303 | JSH-X202 | 3.94 | 1 | Motor: 3.5m×2 Non-motor: 1.5m×2 (Asphalt) | 2170 | 7.5% | Industry |
| | X202-XYQ | 1.11 | 1 | | 1620 | 8.4% | Residence |
| | XYQ-X203 | 3.79 | 1 | | 1540 | 8.2% | Residence |
| | X203 west | 3.99 | 1 | | 640 | 6.2% | Agriculture |
| Jinzhuang (JZ) | X301-X302 | 3.85 | 4 | Motor: 3.5m×2 Non-motor: 1.5m×2 + Motor: 3.5m×2 Lateral barriers: 0.5m×2 Non-motor: 2m×2 (Asphalt) | 1350 | 2.1% | Residence |
| Xiangyangqiao (XYQ) | X303-X203 | 4.87 | 2 | Motor: 3.5m×2 Non-motor: 1.5m×2 (Asphalt) | 780 | 3.6% | Residence and commercial |
| Jincheng (JC) | JSH-BT | 1.48 | 2 | Mortor:3.75m×2 Non-Motor: 1.5m×2 (Cement) | 1230 | 24.5% | Industry and warehouse |
| | BT-NDY | 0.88 | 2 | | 1510 | 27.7% | |
| | NDY-NHY | 1.62 | 2 | | 1670 | 31.6% | |
| | NHY-NJH | 1.71 | 2 | | 1380 | 32.1% | |
| . . . | . . . | . . . | . . . | . . . | . . . | . . . | . . . |
| X204 | X302 north | 1.51 | 1 | Mortor:3.5m×2 Non-Motor: 1.5m×2 (Asphalt) | 560 | 4.1% | Agriculture |
| | X302-S265 | 1.78 | 1 | | 660 | 4.7% | Residence |
| | S265-X303 | 3.11 | 1 | | 590 | 6.2% | Industry and agriculture |
| | X303 south | 7.84 | 1 | | 630 | 5.4% | |
| Jinsha (JSH) | JC south | 1.57 | 2 | Mortor:3.75m×4 Central green belt:2m Non-Motor: 1.5m×2 (Asphalt) | 4310 | 5.9% | Industry and agriculture |
| | JC-NJH | 5.66 | 2 | | 4200 | 5.5% | |
| | NJH-X301 | 1.73 | 2 | | 3780 | 6.4% | |

1. Warning signs
   The current road section has major warning signs such as pedestrian crossings, hazardous locations, and crossings. Children's caution signs are placed above the motor lanes in the area near schools.

2. Prohibition signs
   Speed limit signs are distributed in the places along the town area and the factory area, and the weight limit signs are further installed before bridges to limit vehicles axle weight. No-parking signs are scattered to ensure the right-of-way for non-motorized traffic.

3. Protection infrastructures
   Even though there are barriers between motor lanes and non-motor lanes, guardrails are installed along parts of slopes or rivers.

**Table 2. Illustrations for typical rural road cross-section type.**

| Type No. | Cross-section | Typical lane distribution | Characteristic |
|---|---|---|---|
| 1 | Single carriageway | Motor: 3.5m×2<br>Non-motor: 1.5m×2 | •without central separations;<br>• mixed traffic flow and conflict |
| 2 | Dual carriageway | Motor: 3.5m×4<br>Non-motor: 2m×2<br>Central green belt: 2m | •with central separations;<br>• without directional conflict;<br>• with lateral conflict between motor and non-motor traffic |
| 3 | Triple carriageway | Motor: 3.75m×2<br>Non-motor: 3m×2<br>Lateral separations: 0.75m×2 | •with two lateral separations;<br>• without motor and non-motor conflict |
| 4 | Mixed type | Single carriageway + Dual carriageway<br>Single carriageway + Triple carriageway | The main road is single carriageway, the multi-carriageway appears in the road sections across the town area to improve the traffic safety. |

**X302 Road.** The investigated road section spans from the X302-X203 intersection to the X302-JSH intersection, with a total length of 12.5 km. The investigated section is a 10 meters wide single carriageway without bilateral non-motorized lanes. The basic condition of cross section and related facilities are shown in Fig 7.

1. Warning signs. Because there exists large number of villages near current road, the warning signs of non-signalized T type crossing and village caution are distributed along X302. Meanwhile, in the section with poor visibility and alignment, sings like sharp turn and side road are also set accordingly.

2. Prohibition signs. Most speed limits of X302 are 60 km/h, while in the section passing town area, the speed limits will decrease to 30km/h. Due to the branch ways connecting villages and current road, yield signs have been set on the corner of the branch way. No-parking signs are also laid along the busy town sections.

3. Protection infrastructures. Fluorescent arrow signs and guardrails are not the only safety measures on dangerous roads. Vibration markings and red-white posts are also often distributed. The vibration markings are placed before the bends as a reminder of slowing down, and the red-white reflective posts are used to identify the crossings of branch ways.

**Jianshe Road.** The investigated road section spans from the JS-X302 intersection to the southern provincial road, with a total length of 9 km. The corresponding cross section is similar to the Baita Road. Major signs and facilities are indicated in Fig 8.

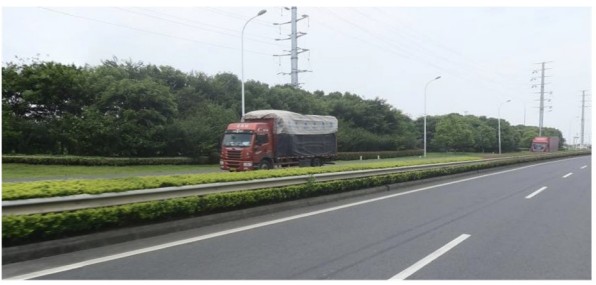
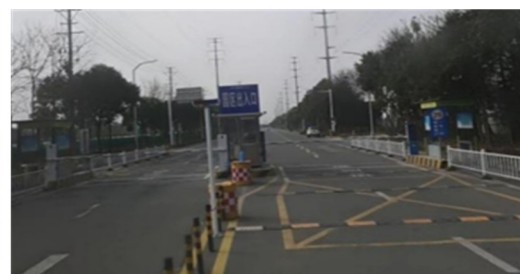

(a) The central green belt on JSH Rd.          (b) The central guardrails on NHY Rd.

**Fig 3. Different isolation forms of dual carriageways.**

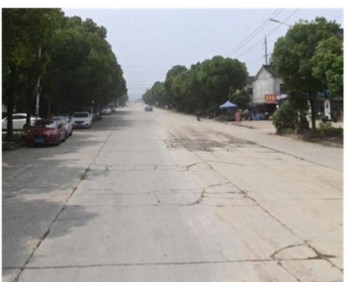 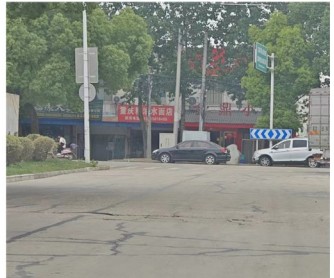 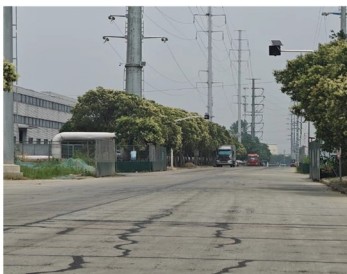

(a) Cracks on NHY Rd.  (b) Cracks on DC Rd.  (c) Cracks on JC Rd.

**Fig 4. Pavement cracks on rural roads with cement pavement.**

1. Warning signs. Compared with the foregoing sections, current road section is another equipped with signs of narrow roads.

2. Prohibition signs. The major prohibition signs on current road include the signs of speed limits, weight limits and the no overtaking, where the no overtaking signs are placed along the curve sections.

3. Protection infrastructures. The current road has a variety of traffic protection infrastructures. Besides the abovementioned side guardrails and surface vibration markings, the yellow flashing lights are deployed to remind drivers slowing down, and the warning lights and message boards are specially set before the adjacent branches to identify the non-signalized crossings.

## Road infrastructure condition analysis

According to the investigated data of rural road sections, the problems and enhancement strategies have been discussed from the perspectives of facility configuration, traffic management and maintenance, and network topology.

**Adopted standards and guidelines.** Two standards are the primary basis for inspecting traffic signs. One is the Road Traffic Signs and Markings (GB5768.2–2022), the national standard applied in China. And the other is the Road Traffic Sign Panels (JT/T 279–2004), issued

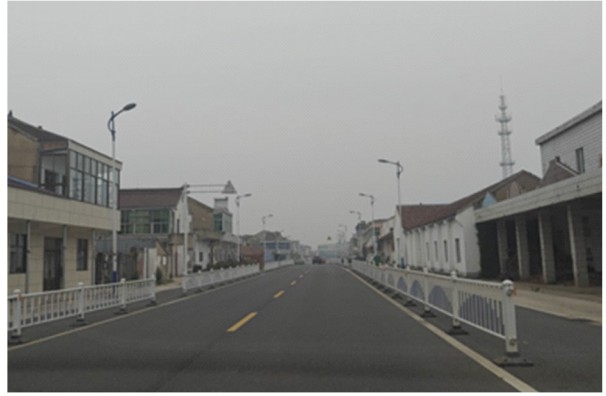 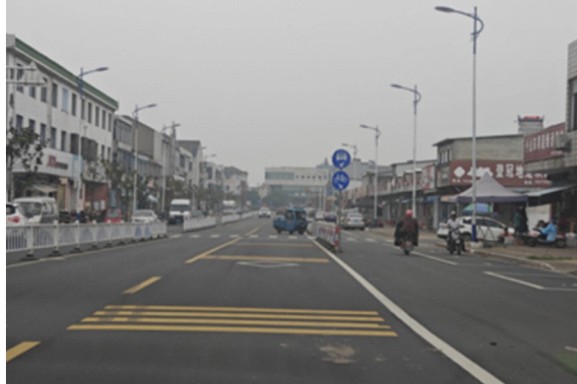

（a）Sections along the JZ Rd.  （b）Sections along the X203 Rd.

**Fig 5. Representative three-carriageway rural road with barriers.**

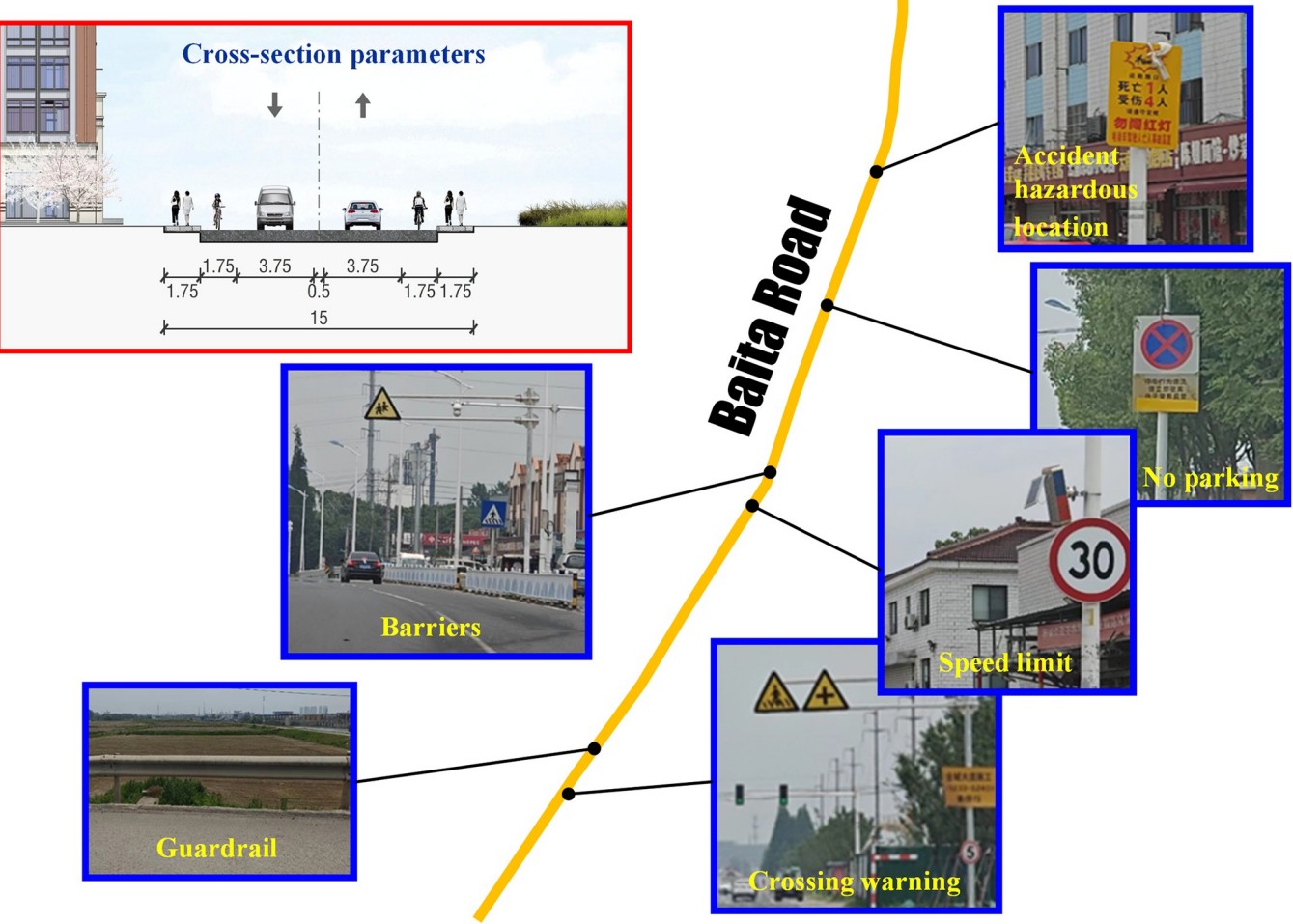

**Fig 6. Major traffic signs and facilities on Biata Road.**

by the Ministry of Transport. Generally, the adopted standards are summarized from the following three perspectives according to our study focus. It should be pointed out that the adopted standards and guidelines for facility inspection are restricted to China, as rural road transportation conditions vary in different countries and regions.

**Quantified standards for sign size and appearance.**

1. The size of sign panels should conform to the related standards in GB5768.2, where the acceptable size deviation is ± 5mm for normal panels, and ± 0.5% for panels longer than 1.2m.

2. Sign panels should have a flat surface with less than 3mm/m unevenness.

3. The following defects are not allowed on the traffic sign panels: cracks, wrinkles and peeling-off edges; obvious scratches, damages and the non-uniform color painting; blisters or bubbles larger than 10 mm2 on any surface of a 50cm×50cm size; the uneven retroreflective performance.

4. The size of sign panels depends on the roads design speed. Table 3 shows the corresponding sizes of different prohibitions and size shapes for rural roads with a design speed between 40 km/h and 70 km/h.

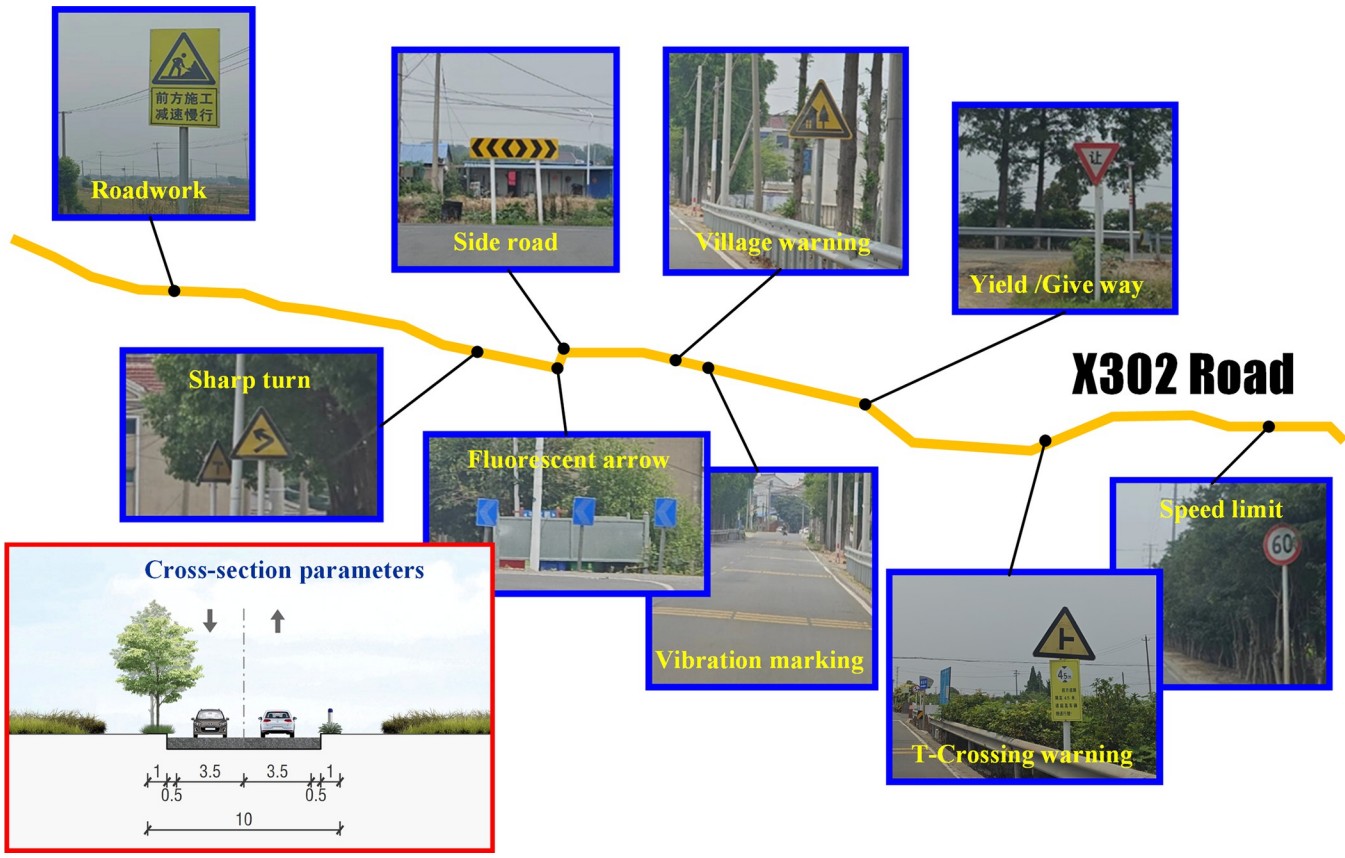

**Fig 7. Major traffic signs and facilities on X302 Road.**

**Requirements for sign angles.**

1. The installation angle should avoid the glare to drivers from the sign panel.

2. It is important that the roadside traffic signs are either vertically aligned with the road centerline or at a specific angle from that direction. Specifically, the angle of prohibition signs and directional signs should be 0~10˚ or 30~45˚, the angle of warning signs and guiding signs should be 0~10˚. As indicated in the Fig 9.

**Regulations for sign locations.** Due to the large number of T-type crossings or non-signalized crossings, special focuses are paid on the yielding signs and branch way posts.

The stop yielding signs should be installed on the mouth of branch way under the following conditions: the left-turn motor traffic needs control, or the conflict between motor traffic and non-motor traffic needs control; the crossing sight distance is limited or the intersection is a hazardous location; the signal intersection without all day 24-hour control.

As compared to the stop yielding signs, the speed reduction yielding signs can be installed at the mouth of branch way with a better sight distance or a lower accident risk, or at the entrances of non-signal roundabouts.

The branch way posts should be installed on the two sides of branch mouth, in order to warn the drivers along the main road against the potential conflict with vehicles from the branch way. The location and feature of branch way posts are illustrated in Fig 10.

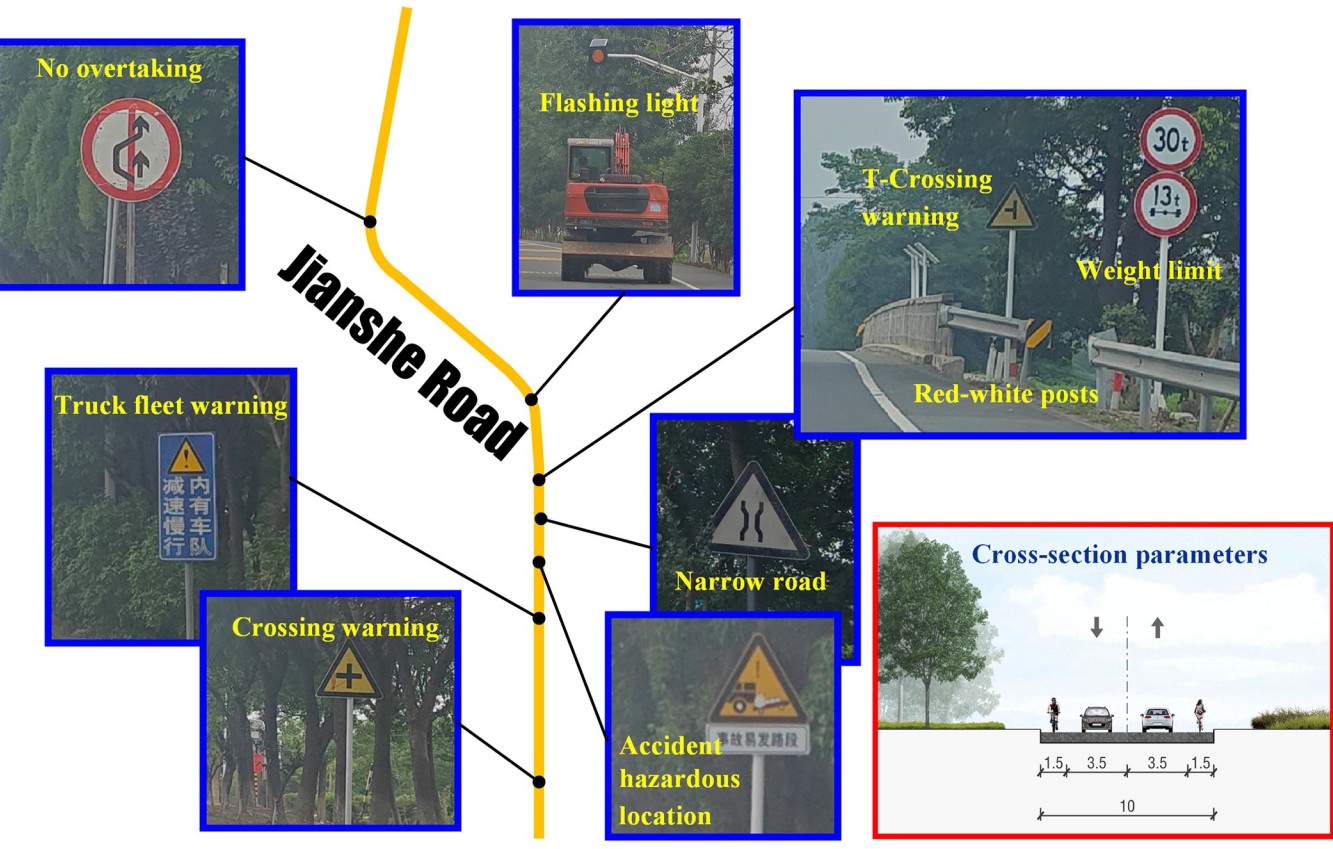

**Fig 8. Major traffic signs and facilities on Jianshe Road.**

## Traffic facility configuration

Despite the fact that the investigated rural roads have equipped with a variety of warning signs and prohibition signs, there do exist obvious defects in the aspects of sign type, install location and timely update.

**Traffic signs types.** The following traffic signs are missed along the road sections considering the surrounding landscapes and traffic characteristics, as shown in Fig 11.

1. Speed limit release signs. The top speed of rural roads is 60 km/h, and the speed will be limited to 30 km/h or 20km/h in the accident hazardous sections and the town area sections under the roadside warning signs, but the speed limit release signs are seldom set in the

**Table 3. Standard shape sizes for typical signs in rural area.**

| Shape type | | Parameters | Size value (cm) |
|---|---|---|---|
| Octagonal sign | | Outside diameter | 80~100 |
| Triangular sign | | Side length | 90~110 |
| Circular sign | | Outside diameter | 80 |
| Rectangle sign | Off-limits or restricted area sign | Height×Width | 170×120 |
| | One-way traffic sign | Height×Width | 80×40 |
| | Square sign | Outside diameter | 80 |
| | Other rectangle signs | Height×Width | 80×64 |

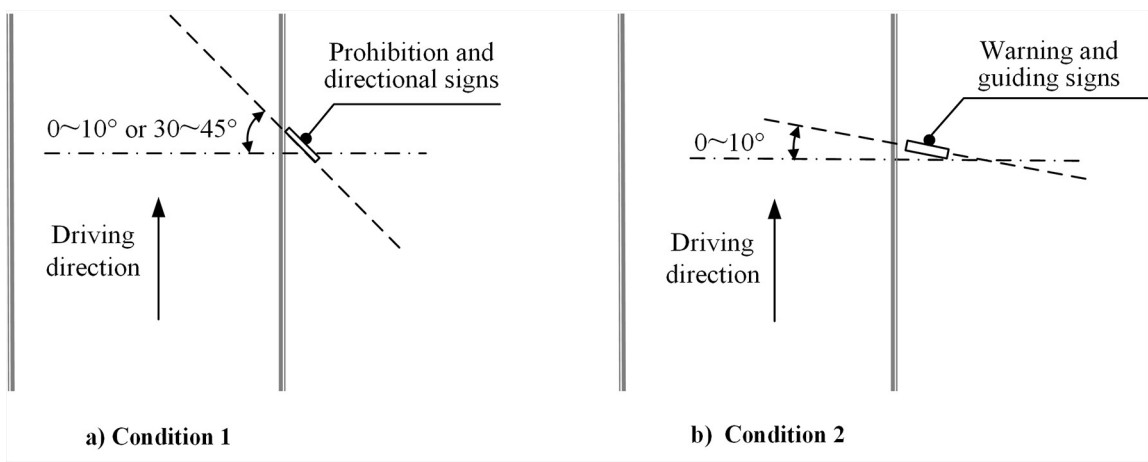

**Fig 9. The requirements for the angles of specific traffic signs.**

investigated road net. It is necessary to tell drivers the boundary of speed limit section, so that the drivers can decide when to accelerate to save the trip time.

2. No tooting signs. In the road sections passing through the street community and town market, no tooting signs should be placed appropriately in order to facilitate the construction of a peaceful and harmonious rural community [20].

3. Continuous curve signs. For those adjacent road sections with sharp or big curves, the single sharp turn signs should be updated to the continuous curve signs, in order to inform the drivers of poor alignments ahead.

4. Livestock caution signs. During the investigation, it is found that some sheep or geese groups will cross the rural road, while the traffic speed is relatively higher. Therefore, it is indispensable to set the livestock caution signs in the related section, together with the signs of speed limit at 30 km/h.

5. Other warning signs. Other warning signs including the Y-type crossings, staggered crossings and hump bridges should also be installed in the appropriate road section.

**Install location.** The problems of traffic related signs and facilities location are:

1. Guardrail missing. The guardrails mainly refer to the roadside guardrails. Some road sections along the river have not installed the guardrail due to the existence of trees or barriers.

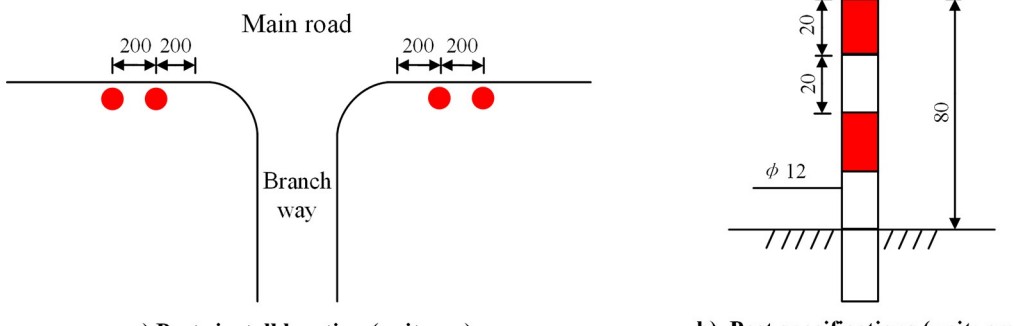

**Fig 10. The installation standards of branch way posts.**

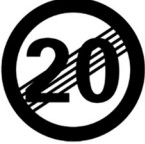 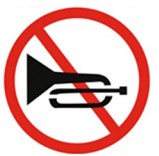 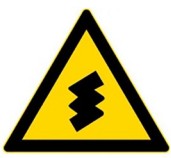 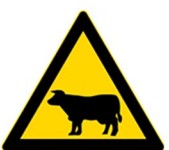 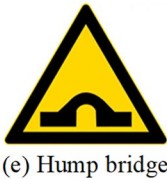

(a) Speed limit release sign | (b) No tooting sign | (c) Continuous curve sign | (d) Livestock caution | (e) Hump bridge sign

**Fig 11. Necessary signs missed on current rural roads.**

2. Occluded signs. Along the rural roads, a great many of the signs are occluded by the trees, buildings and telegraph poles, and some sign columns are set to a wrong angle, which will affect the perception and action of drivers. Therefore, it is suggested to clear the surrounding area of current occluded signs, or set the signs in a more apparent location without occlusion.

3. Overlapped signs. In some special area like bridges and streets, the phenomenon that several kinds of warning signs and prohibition signs are installed together, namely the information are displayed too intensive, which is adverse for drivers to perceive information effectively.

**Facility updates and maintenance.**   In the rural area, traffic safety related facilities lack timely updates and maintenance, mainly reflected in the following aspects.

1. Different degrees of damage and fade of traffic signs. The fade of yellow warning signs will affect the night visibility, and the common damage status include the tilted column, the folded or bent sign and the worn surface upon our investigation. According to the observation data, the reflective performance of faded signs has decreased by 15% to 40%.

2. Unclear road surface markings. Under the influence of road cracks, some motor lane lines and center lines are worn away to varying degrees, making it harder to distinguish the solid lines and the dotted lines. The aggregate length of road sections with unclear markings is about 6.1 km.

3. Broken branch way posts. Some red-white posts used to identify branch ways are broken, and some posts lost the reflective coatings. According to the regulations for branch way posts, nearly 1/3 non-signal control T-type crossings are found disqualified both in the number of posts and in the height of posts.

4. Bended guardrails, Along the section with adjacent rivers or lakes, some roadside guardrails have been hit by vehicles and become bended for a long while. It is suggested to inspect the status of guardrails and conduct timely maintenance.

## Traffic management and maintenance

The following three kinds of traffic control risks are found during our investigation.

1. Trucks traffic management
   The rural area being studied has several industry parks and factories that produce energy, chemical, and building materials, which leads to a significant demand for truck traffic. On one hand, the phenomenon of trucks illegal parking has occupied some road sections, making it difficult for vehicles to meet. On the other hand, the heavy trucks have caused severe

problems of road surface cracks [21] and fugitive dusts, especially on the cement paved roads. Hence, it is significant to enhance the management of trucks by setting parking rules and planning driving routes.

2. Non-signalized crossings control
   The non-signalized crossings along the rural roads mainly include the T type crossings and the Y type crossings under the priority control rules, by setting the yield signs on branch ways. Due to the presence of lush shrubs and trees along the roads, the sight triangle was unable to satisfy the driving visibility demand, resulting in merging conflicts at the intersection.

3. Roadworks management
   During the investigation period, some road sections are being rebuilt or maintained, and the necessary management measures need to be enhanced. In the micro section traffic safety organization, the current measure is just place speed limit signs and warning signs before the work area, without signs of narrow road or lane blockage, nor the protection facilities like anti-collision buckets and cones. At the global network level, there lacks bulletin boards displaying roadwork information and path guidance suggestion on the adjacent road, which may increase the detour distance of some vehicles.

4. Road surface maintenance
   Due to the repeated demand of heavy trucks, roads around the industry parks and factories have suffered from surface cracks, as indicated in Fig 4. The cracks will affect the driving stability and comfortability, especially threatening the safety of non-motorized vehicles on the single carriageway section.

## Rural roadnet connectivity

The relation between road accessibility and rural economics has been verified in current studies [22]. In order to analyze the roadnet connectivity of studied area, the theory of complex network is applied here to calculate the corresponding indicators, mainly including the degree centrality, the clustering coefficient and the average shortest path length [23]. The results of complex network analysis will support the evaluation of the level of service and safety. The undirected network is output upon the adjacency matrix of the actual roadnet, as indicated in Fig 12.

**Degree centrality.** To describe the connectivity between the current node and it's adjacent nodes, the indicator of degree centrality is used. The degree of node $i$ is defined by the number of its surrounding edges $ki$. E.g., the degree of an intersection point is 4, the degree of a T-type crossing is 3, and the degree of a dead-end road section is 1.

The node degree distribution is shown in Fig 13(A). The node degree in the current network ranges from 1 to 4, with an average node degree of 2.7, which approaches the degree of T-type or Y-type crossings, validating that there exist many T-type crossings in the area.

**Clustering coefficient.** The clustering coefficient is used to describe the clustering effect of current node, and is defined by the actual number of edges between nodes adjacent to current node, as well as the possible number of edges between the adjacent nodes, as indicated in Eq (1).

$$C_i = \frac{E_i}{k_i \times (k_i - 1)} \tag{1}$$

Where $C_i$ refers to the clustering coefficient of node $i$; $E_i$ is the actual number of edges connecting the nodes adjacent to current node $i$; $k_i$ is the degree of node $i$.

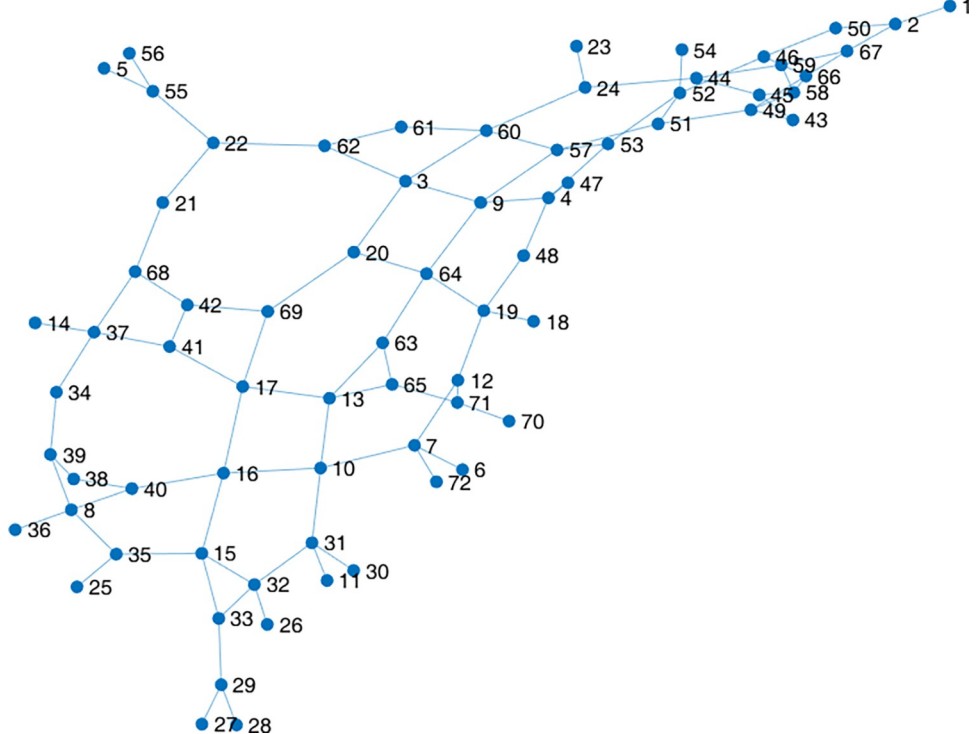

**Fig 12. The undirected network topology of studied rural roadnet.**

Via self-coded batch processing algorithm, the clustering coefficient of 72 nodes are indicated in Fig 13(B). It is obvious that most of the surrounding nodes are not connected to each other. Only 9 nodes' clustering coefficient are above zero, and the average clustering coefficient is about 0.028. The connection levels of the overall network nodes are inferior to those of the nodes in the urban roadnet.

**Average length of shortest paths.** The average length of shortest paths is the average value of shortest paths between all nodes, reflecting the accessibility inside the network. The average length is calculated by Eq (2). The length in the complex network is a dimensionless parameter, it is not equal to the actual road length. Namely, the length of one edge connecting

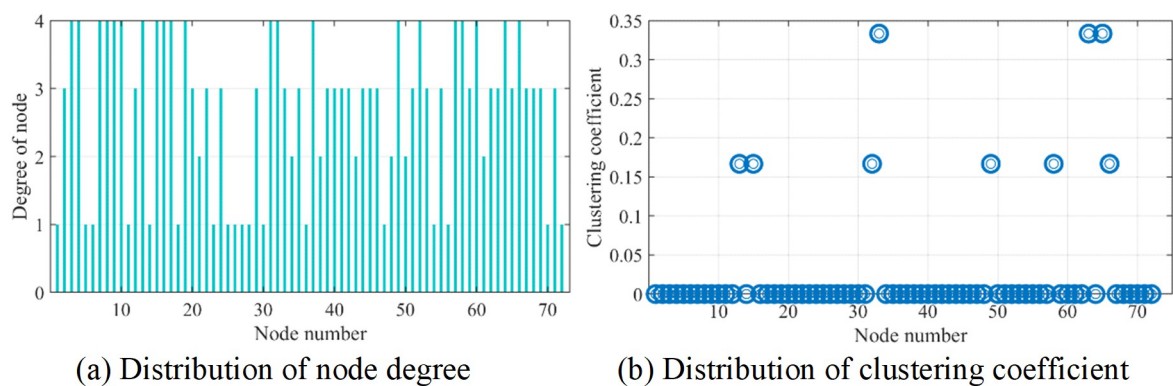

(a) Distribution of node degree    (b) Distribution of clustering coefficient

**Fig 13. The degree and clustering coefficient of network nodes.**

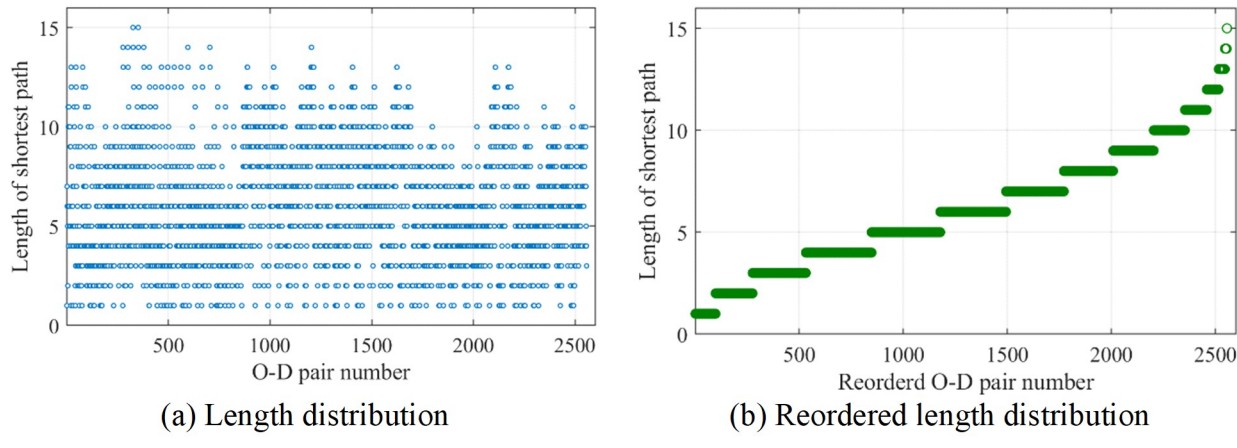

(a) Length distribution                                            (b) Reordered length distribution

**Fig 14. The length distribution of shortest paths.**

adjacent nodes is 1, and one path is composed of several edges.

$$L = \sum_{i \geq j}^{N} d_{ij}/[N \times (N-1)/2] \quad i, j \in [1, N] \tag{2}$$

where $L$ refers to the average length of network shortest paths; $d_{ij}$ is the shortest path length between node $i$ and node $j$, generated by the Floyd-Warshall algorithm [24]; $N$ is the number of total nodes in the complex network.

Within the current network, there are 72 nodes, corresponding to 2556 ($C_{72}^2$) origin-destination pairs. The lengths of these 2556 shortest path are illustrated in Fig 14(A). For the convenience, the lengths of shortest paths have been reordered ascendingly, as indicated in Fig 14(B). Apparently, the shortest path lengths range from 1 to 15, 86% of the path lengths are less than 9, and lengths of 4 to 7 account for about 50%. The average length of all shortest paths is about 6.07, meaning that travel from one node to any another node will pass through 6.07 edges in average.

In order to analyze the connectivity level, we have performed a comparison analysis between the studied area and the rural areas with high-quality roadnet construction, as indicated in Table 4. The average node degree of the studied area is slightly higher than the Feixi district, while lower than the Huzhou and Enshi district, because the latter two rural districts own more signalized intersections. Meanwhile, the studied area underperforms in the other two indicators, indicating that the local accessibility and connectivity need improvement.

**Table 4. Key indicators comparison among rural road network.**

| District | Average degree | Average clustering coefficient | Average length of shortest paths |
|---|---|---|---|
| Huzhou, Zhejiang Province | 3.2 | 0.051 | 5.18 |
| Enshi, Hubei Province | 3.1 | 0.037 | 5.63 |
| Feixi, Anhui Province | 2.6 | 0.035 | 5.95 |
| Studied Area | 2.7 | 0.028 | 6.07 |

## Enhancement mechanism for rural roads infrastructure

**Enhancement rules.** Targeting at the high-quality development of rural roads, the following four rules have been put forward in order to achieve the status where drivers enjoy their trips and goods enjoy their circulations.

1. Classified management. The management scheme should consider the differences among rural roads with different conditions and realize a problem-oriented enhancement.

2. Symbiotic development. Since the rural roads major serve the demands of passenger and freight traffic in the rural districts, the enhancement or reconstruction should merge into the development of ecological country or tourist town, providing coordinated infrastructures and facilities for local development.

3. Multi-pronged measures. In order to enhance globally, it is necessary to incorporate road traffic safety, network connectivity, and service sustainability measures, while also focusing on the integrated lifecycle development of planning, construction, management, and maintenance.

4. Clear liabilities. It is of urgent need to improve the liability system of rural road management, where the duties of local government, transport management department, law-enforcing department and agricultural sector should be clarified hierarchically.

## Enhancement strategies

**Quantified traffic safety inspection.** The result of field observation and measurement shows that current configurations of traffic facilities on rural roads are not optimistic, lagging behind the national and industrial standards. In order to scientifically assess the level of road service and safety, an assessment framework is established considering the indicators from perspective of road alignment infrastructure, roadside landscape, traffic condition, traffic signs and protection facilities, as indicated in Tables 5 and 6, where the road alignment, roadside landscape and traffic condition belong to the auxiliary indicators, and the others are categorized as the fundamental indicators.

**Traffic control enhancement.** According to the investigation results, rural road traffic is more susceptible to uncertainty and illegal activity than urban road traffic. E.g., over-speeding behaviors occur frequently on low-volume sections according to aerial video analysis, which makes crossings dangerous. The traffic control of rural roads can be enhanced from the following perspectives.

(1) Signalized crossings setting. With the development of rural economy, the traffic volume will arise accordingly. Based on the particular investigation on network traffic volumes distribution, some non-signalized intersections with higher volumes should be updated as signal control crossings to reduce the merging conflicts, where the decision of timing scheme should consider the dynamicity of volumes temporal distribution.

(2) Visibility enhancement for non-signalized crossings. As shown in Fig 15, the sight triangle should both consider the sight point, sight boundary and sight angle, where the sight boundary is the axis of the outer lane, and the sight angle is 60°. The sight distance $S_1$ and $S_2$ are calculated by Eq (3) and Eq (4) respectively.

$$S_1 = (n_l + 0.5)w_l + w_{rs} + w_{bs} \qquad (3)$$

**Table 5. The quantifiable auxiliary indicators related with traffic safety.**

| Aspect | Indicator | Unit | Notes |
|---|---|---|---|
| Road alignment | Section length | km | / |
| | Number of sharp turn | / | Horizontal radius < 50 m |
| | Number of continuous curve | / | Excluding single sharp turns |
| | Ramp/Slope length | km | / |
| | Number of carriageways | / | / |
| | Number of poor visibility crossings | / | Determined by the sight triangle |
| | Number of narrow road sections | / | Including the section under reconstruction |
| | Lane width | m | / |
| Roadside landscape | Number of villages | / | / |
| | Number of occluded signs | / | Occluded by trees or shrubs |
| | Length of waterside sections | km | Along the river or lakes |
| | Population density | person/km$^2$ | Population lived in the roadside villages |
| | Number of crossed branch ways | / | / |
| Traffic conditions | Traffic volume | PCU/day | PCU is the passenger car unit |
| | Average speed | km/h | Considering different vehicle types |
| | Ratio of trucks | % | / |
| | Cross-road volume | person/day | Including pedestrian and non-motorized vehicles |
| | Accident frequency | times/year | Provided by the traffic sector |
| | Average accident economic lost | RMB | Measured by the direct economic losses |

**Table 6. The fundamental items related with traffic safety.**

| Aspect | Items | Notes |
|---|---|---|
| Prohibition signs | Number of normal speed limit | The number of each sign kind is composed of three parts:<br>• the number of signs meeting the issued standards;<br>• the number of signs failing to meet the issued standards in the location, shape size, height, angle or the visibility;<br>• the number of missing signs according to the guidelines. |
| | Number of special speed limit | |
| | Number of no parking signs | |
| | Number of no overtaking signs | |
| Warning signs | Number of intersection signs | |
| | Number of T-type crossing signs | |
| | Number of village signs | |
| | Number of sharp turn signs | |
| | Number of narrow road signs | |
| | Number of yield signs on branches | Including the speed reduction yielding and the stop yielding |
| Protection facilities | Guardrail length | Unit: m |
| | Isolation barriers length | Unit: m |
| | Length of speed reduction markings | Unit: m |
| | Number of branch way posts | Unit: pair |
| | Number of flashing lights | Including the yellow flashing and the red-white flashing |

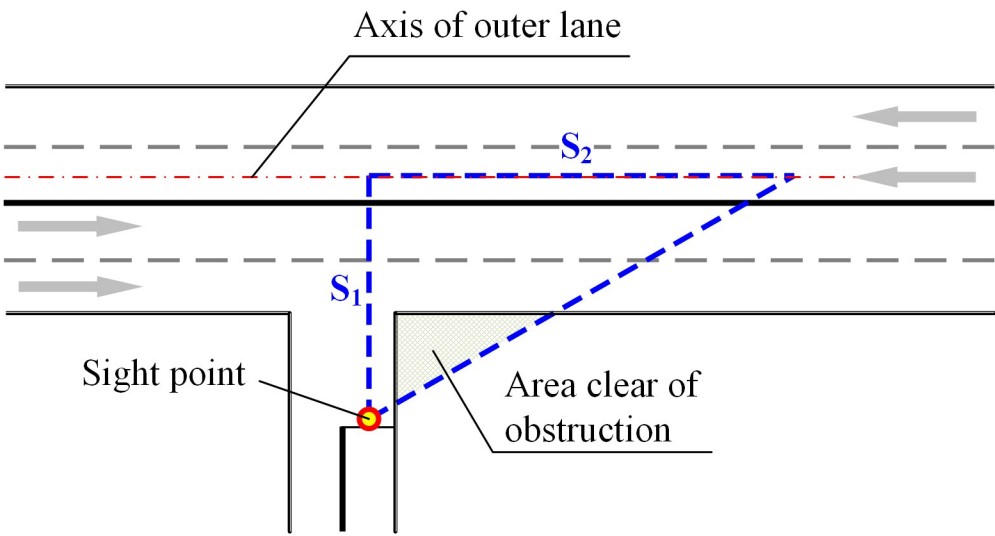

**Fig 15. The sight triangle at the T-crossing of branch way and the main road.**

$$S_2 = \sqrt{3}S_1 \tag{4}$$

where $n_l$ is number of motor lanes of one direction, $w_l$ is the lane width, $w_{rs}$ is the remaining width of non-motor lane and road shoulder, and $w_{bs}$ is the safety distance before the stop line.

Taking the crossing of a two-carriageways road and a single-carriageway branch, the sight distance $S_1$ should be at least 11.75 meters, $S_2$ should be 20.4 m. Meanwhile, the shrubs or structures inside the clear area should be controlled under 1.2 m, according to the *Specification for design of intersections* (CJJ152-2010) published by the Ministry of Housing and Urban-Rural Development of China.

(3) Truck management. Based on the location of industries and factories, it is necessary to plan the regular truck driving paths on the rural network. Guardrails or barriers are required on the road sections of trucks driving paths, and asphalt pavement is required. Meanwhile, the parking area and no-tooting sections should also be planned in advance.

(4) Roadwork management. To minimize conflicts in the construction area, traffic flow organizations near the roadwork area should pay attention to the occupancy status of the lanes [25]. Taking the roadwork occupying half range of a 4-lane road as an example, the local distribution of traffic signs and facilities should be roadwork warning sign, speed limit sign, narrow road sign, anti-collision facilities and speed release signs in sequence. If the road section is full occupied, besides the warning signs of road blockage and no passing, regional paths guiding boards are suggested to install around the area to realize the microsimulation of network flows, as indicated in Fig 16(A) and 16(b).

**Intelligent management system for rural roads.** In order to effectively enhance the performance of rural road network and conduct real-time control strategies, a framework of intelligent management system is presented in this section, as indicated in Fig 17.

The system is composed of the following basic modules.

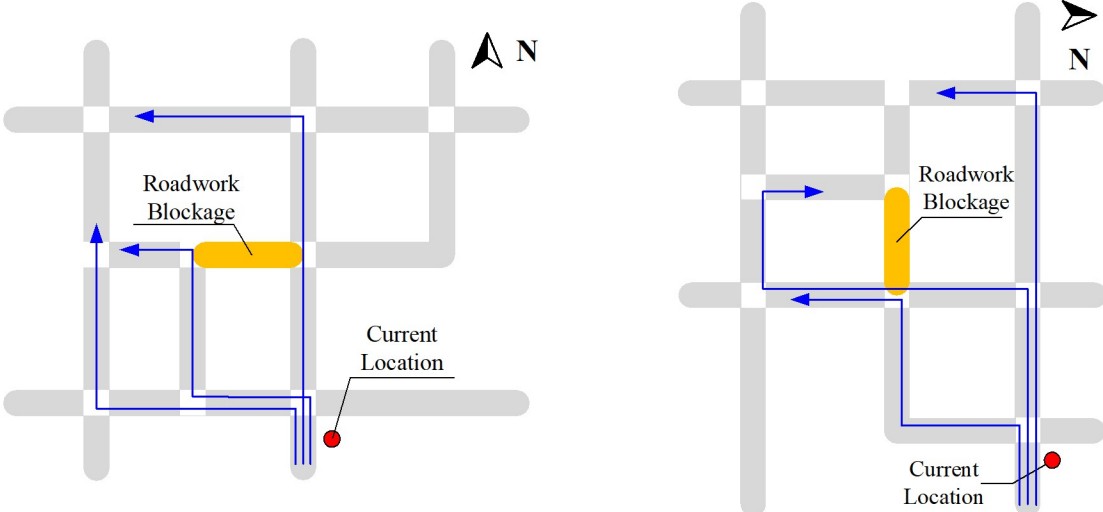

(a) Boards at southeast intersection    (b) Boards at northeast intersection

**Fig 16. Path guidance at different locations around blockage area.**

(1) Data acquisition module. The data acquisition module is an integrated database of video-based data, the sensor-based data and the historical data. The video-based data include the traffic volume, vehicle velocity and traffic density, which can be extracted from the video upon mature algorithms and theories of traffic-related image recognition [26]. The video data can be provided by the fixed cameras and the unmanned aerial vehicles. The sensor-based data include the traffic noise, vehicles emission and truckloads. The historical data include traffic accidents, maintenance record, pavement roughness, GIS data and etc., where the pavement roughness data are collected by periodic laser scanning.

(2) Risk identification module. Based on the data of historical traffic accident, flow characteristics and road infrastructures, current module is designed to identify the road sections risk distribution, and find out locations with poor safety status. The use of hierarchical analysis or fuzzy clustering method can help identify the safety status, allowing for better daily management and regular inspection. As indicated in Fig 17, the global risk distribution of rural road network is reflected by a heat map, where a deeper color means a higher severity of road sections, with a higher possibility of traffic conflicts or a worse condition of road infrastructures. The risk severity of the section and the crossing are quantified by Eq (5) and Eq (6) respectively. Results show that over 35% of road sections and crossings in the town area suffer from higher risk perturbations.

$$R_{\mathrm{se}} = P_{\mathrm{se}} \left( \frac{V_{\mathrm{se}}}{Cap_{\mathrm{se}}} \right)^{1/2} \times \left( 1 + \frac{PCE_{\mathrm{tr}} V_{\mathrm{tr}}}{V_{\mathrm{se}}} + \frac{L_{\mathrm{da}}}{L_{\mathrm{se}}} \right) \tag{5}$$

$$R_{\mathrm{cr}} = \begin{cases} P_{\mathrm{cr}} & \text{Signalized control} \\ P_{\mathrm{cr}} \times \left[ 1 + (1 + \cos\alpha) \dfrac{V_{\mathrm{br}}^{\mathrm{le}} + V_{\mathrm{mr}}^{\mathrm{le}}}{V_{\mathrm{cr}}} \right] & \text{Non−signalized control} \end{cases} \tag{6}$$

In Eq (5), $R_{se}$ refers to the section risk severity; $P_{se}$ denotes the basic accident probability on current road section; $V_{se}$ is the daily traffic volume from 8:00 am to 9:00 pm, PCU/day; $Cap_{se}$ is

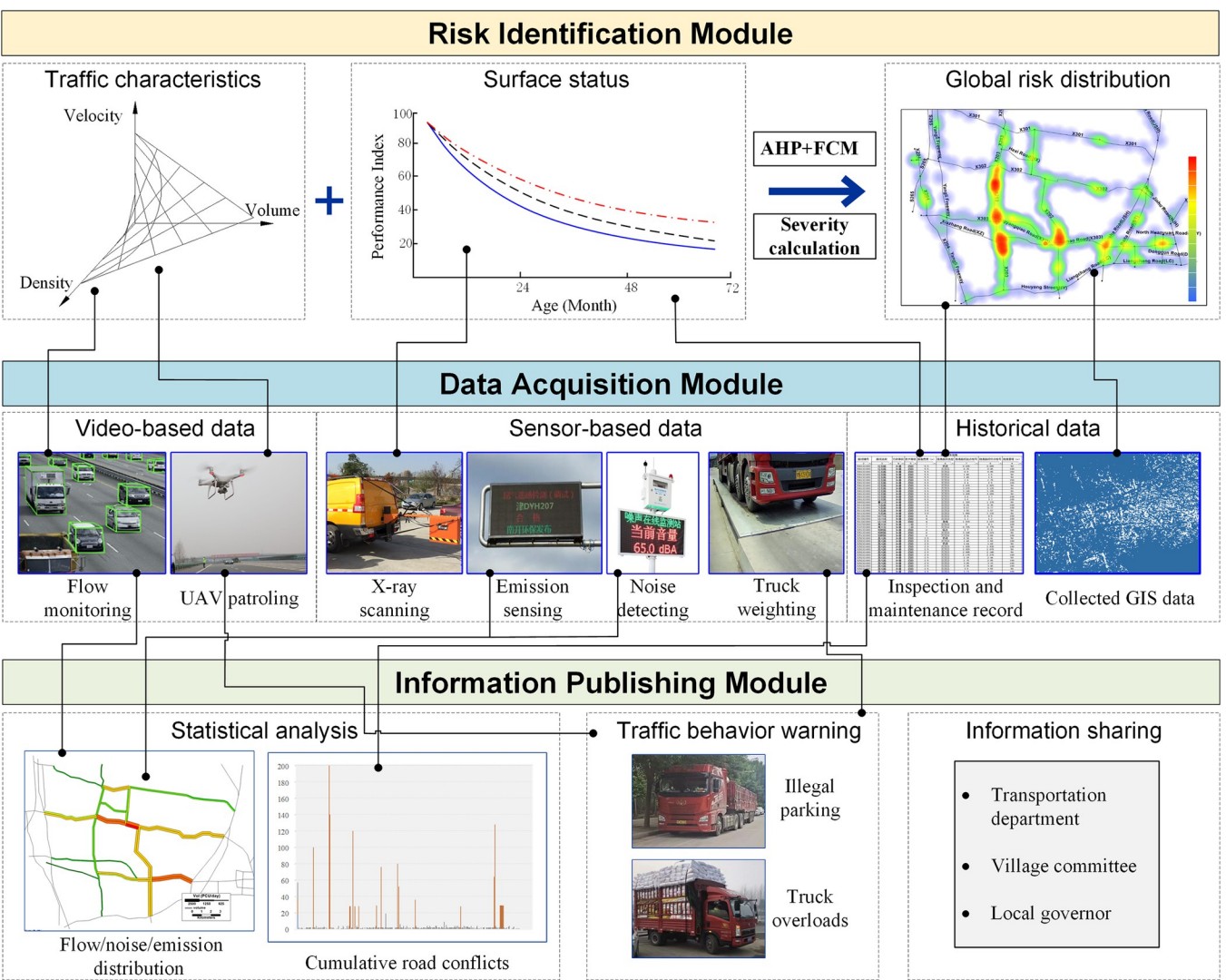

**Fig 17. Propose framework for rural road management system.** (The two maps are republished from TransCAD under a CC BY license, with permission from Caliper Corporation, original copyright 2024).

the corresponding lane capacity, PCU/day, affected by the lane width, cross-section type and speed limit; $V_{tr}$ is the volume of trucks, veh/day, and $PCE_{tr}$ is the passenger car equivalent of trucks used to convert $V_{tr}$ to standard volume, usually takes the value of 4; $L_{da}$ is the length of damaged road surface including cracks and depressions, km; $L_{se}$ is the length of current road section, km.

In Eq (6), the calculation of crossing risk $R_{cr}$ is influenced by the control type. For the signalized crossings, the risk can be equivalent to $P_{cr}$, the basic accident probability around current sections. For the non-signalized control T-type or Y-type crossings, the risk should be modified by the crossing angle and the conflicted volume, where $\alpha$ denotes the smaller angle between the main road and the branch way, a smaller $\alpha$ means a worse visibility. $V_{br}^{le}$ is the left-turning volume of vehicles on the branch way, PCU/day; $V_{mr}^{le}$ is the left-turning volume of vehicles on the main road, PCU/day; $V_{cr}$ is the volume of the crossings, PCU/day, compose of the bi-directional volumes on both main road and branch way.

(3) Information publishing module. The purpose of this module is to quantitatively analyze and visualize important data, including real-time network accessibility, traffic flow, emission, noise distribution, illegal traffic behavior, and other relevant information. Meanwhile, the published information should be shared among transportation departments, village committees and local governors, for the convenience of coordinated administration.

## Conclusions

This main contribution of this paper lies in improving the level of service and safety for rural roads through the following two aspects. The first is the evaluation methodology for rural roads infrastructures, composed of field investigation, traffic facility configuration, management and maintenance, and network connectivity. The field study reveals that the common problems of rural roads are safety infrastructure maintenance and traffic operation management, arriving from the outdated construction standards and the long-term absence of effective supervision, e.g. the number of effective warning signs only accounts for 69% according to the standards. The second is the enhancement mechanism, including the evaluation tables composed of quantifiable indicators, the specific strategies from local traffic enhancement to global improvement, and the necessary modules for the establishment of intelligent management system. The proposed framework of rural roads management has integrated technologies of multi-source data analysis, risk visualization and assistant decision. Although every strategy is feasible and reliable under current theory and technology, some strategy can only be implemented by the combined efforts of relevant engineers, administrators and managers.

Meanwhile, during our analysis and evaluation, it is found that there exists a strong relation between the traffic characteristics and the level of service and safety, e.g., the higher proportion of trucks and lorries usually corresponds to a worse pavement condition, a worse signs and markings visibility and a higher traffic conflict rate. Therefore, it is of great urgency to establish the intelligent management system for rural roads, with embedded technologies of multi-source data analysis, risk visualization and assistant decision.

Current research is our first step into the field of rural roads management. Our future research will be focused on the emergency management of rural roads under adverse weather conditions [27, 28], such as the heavy rainfall and the snowstorm, which will cause the disruption or blockage of partial road sections. The specific studies include the road network resilience evaluation and the paths reorganization algorithm, in order to realize a safe and efficient traffic round-the-clock.

## Acknowledgments

The authors deeply thank professorate senior engineer Wei Shen in Chanzhou City Planning and Design Institute for his assistance in field data acquisition.

## Author Contributions

**Conceptualization:** Qiannan Ai.

**Data curation:** Qiannan Ai.

**Formal analysis:** Yuling Ye.

**Funding acquisition:** Jun Zhang.

**Investigation:** Qiannan Ai.

**Methodology:** Jun Zhang, Yuling Ye.

**Validation:** Yuling Ye.

**Visualization:** Jun Zhang.

**Writing – original draft:** Qiannan Ai.

**Writing – review & editing:** Jun Zhang, Yuling Ye.

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
