## [Decision Letter · Decision Letter 0]

20 Sep 2023

PONE-D-23-25833Enhancement strategies for Rural Roads management: a case study of Jintan, ChinaPLOS ONE

Dear Dr. Zhang,

Thank you for submitting your manuscript to PLOS ONE. After careful consideration, we feel that it has merit but does not fully meet PLOS ONE’s publication criteria as it currently stands. Therefore, we invite you to submit a revised version of the manuscript that addresses the points raised during the review process.

After careful evaluation and review by our expert reviewers, we have decided to accept your paper pending a major revision. We believe that with substantial improvements addressing the reviewers' comments, your paper has the potential to make a valuable contribution to the academic literature.

Based on the reviewers' comments, we kindly request that you thoroughly revise your paper to address these issues. Please consider incorporating the following key points:

[1] Revise the title to better reflect the content of the paper, aligning it with the strategies to enhance the level of service and safety of rural roads discussed in the study.

[2] Reorganize the literature review, summarizing the relevant points from existing studies and identifying research gaps.

[3] Quantify and provide references for terms and information related to road sections, such as single lane, triple lane, and dimensions, ensuring consistency and clarity.

[4] Improve the presentation of field study observations by quantifying and providing standardized measurements, such as size, shape, and reflectivity, according to applicable guidelines or standards.

[5] Address notational completeness, providing clear explanations and references for equations and tables.

[6] Clarify units of measurement in tables, ensuring consistency and accuracy.

[7] Strengthen the conclusions by highlighting the novel aspects of your work and its contribution to the academic literature.

[8] Enhance the quality of visuals, such as pictures and charts, ensuring they are of sufficient resolution and clarity.

[9] Clearly explain the research methodology employed in your study, including the data collection methods, sample selection, and any statistical analyses performed. Justify how this methodology supports the strategies proposed.

[10] Provide a detailed explanation of the data used in your study, including its sources, relevance, and reliability. Highlight how this data supports and strengthens the suggested strategies.

[11] Consider any limitations or potential biases in your research methodology or data and discuss how these were addressed or mitigated.

We look forward to receiving your revised manuscript.

Kind regards,

Ibrahim Badi, PhD

Academic Editor

PLOS ONE

Journal Requirements:

4. Please ensure that you include a title page within your main document. We do appreciate that you have a title page document uploaded as a separate file, however, as per our author guidelines (http://journals.plos.org/plosone/s/submission-guidelines#loc-title-page) we do require this to be part of the manuscript file itself and not uploaded separately.

   "This work was supported by the Local Innovation Talent Project of Yangzhou under Grant number 2022YXBS118. The authors deeply thank professorate senior engineer Wei Shen in Chanzhou City Planning and Design Institute for his assistance in field data acquisition."

   "Local Innovation Talent Project of Yangzhou under Grant number 2022YXBS118.

Funder: Yangzhou Government, Jiangsu Province, China.

Recipient: Jun Zhang"

6. Thank you for stating the following financial disclosure: 

   "Local Innovation Talent Project of Yangzhou under Grant number 2022YXBS118.

Funder: Yangzhou Government, Jiangsu Province, China.

Recipient: Jun Zhang"

7. We note that you have indicated that data from this study are available upon request. PLOS only allows data to be available upon request if there are legal or ethical restrictions on sharing data publicly. For more information on unacceptable data access restrictions, please see http://journals.plos.org/plosone/s/data-availability#loc-unacceptable-data-access-restrictions. 

8. We note that Figures 1,5,6,7 and 14 in your submission contain map/satellite images which may be copyrighted. All PLOS content is published under the Creative Commons Attribution License (CC BY 4.0), which means that the manuscript, images, and Supporting Information files will be freely available online, and any third party is permitted to access, download, copy, distribute, and use these materials in any way, even commercially, with proper attribution. For these reasons, we cannot publish previously copyrighted maps or satellite images created using proprietary data, such as Google software (Google Maps, Street View, and Earth). For more information, see our copyright guidelines: http://journals.plos.org/plosone/s/licenses-and-copyright.

a. You may seek permission from the original copyright holder of 1,5,6,7 and 14 to publish the content specifically under the CC BY 4.0 license.  

9. We note that Figures 2-4 in your submission contain copyrighted images. All PLOS content is published under the Creative Commons Attribution License (CC BY 4.0), which means that the manuscript, images, and Supporting Information files will be freely available online, and any third party is permitted to access, download, copy, distribute, and use these materials in any way, even commercially, with proper attribution. For more information, see our copyright guidelines: http://journals.plos.org/plosone/s/licenses-and-copyright.

a. You may seek permission from the original copyright holder of Figures 2-4 to publish the content specifically under the CC BY 4.0 license. 

Reviewers' comments:

Reviewer's Responses to Questions

**Comments to the Author**

1. Is the manuscript technically sound, and do the data support the conclusions?

Reviewer #1: No

Reviewer #2: Partly

2. Has the statistical analysis been performed appropriately and rigorously? 

Reviewer #1: No

Reviewer #2: No

3. Have the authors made all data underlying the findings in their manuscript fully available?

Reviewer #1: Yes

Reviewer #2: No

4. Is the manuscript presented in an intelligible fashion and written in standard English?

Reviewer #1: No

Reviewer #2: No

5. Review Comments to the Author

Reviewer #1: In general, the paper could use more editing because there are many places where the wording can be improved. The paper has no valuable contribution to the academic literature. The research methodology and data do not clearly support the suggested strategies.

Reviewer #2: The title of the paper is not communicating the content of the paper" The title can be modified as follows

Strategies to enhance the level of service and safety of a Rural Roads: A Case study.

The literature is not organized properly. Example some where it is written as Tian et al. (2018) and also Griselda et al. pointed out that 49 the completeness and correctness of markings or signs plays an important role in

50 reducing crashes on rural highway [4]. please follow the consistency in writing the same. It is suggested that a good number of studies are available on pavement management system of rural roads or strategies and safety evaluation of rural roads. Further the authors referred many studies, but did not summarize the salient points or takeaway from the existing literature with respect to the current investigation. Also, the research gaps in the literature is missing.

The Table 1 titled with "Basic information of major road sections" i think this is road inventory information!! What do you mean by single or single +triple or dual!! The author is suggested to quantify the same. Some where it is mentioned in the manuscript that single lane means (10m), is there any reference for this!!

All around the world including the developed countries have an 3.75 m as single lane width of the road!! please clarify on the same. !! Similarly what is triple lane ? Authors are suggested to include the road inventory data on such as length of road, width of road , existing pavement cross section details, land use, land pattern, rainfall and traffic data etc.!!

Further, the authors has done a field study and made the observations on the sign boards and other facilities visually! Rather mentioning that the color was faded, the sign boards are not visible and they are not properly protected.....they should have done through quantification in terms of their size, shape and refelctivity as per some standard and the results would have been presented!! for example it is mentioned that " Due to the branch ways connecting villages and current road, yield signs have been set on the169 corner of the branch way. No-parking signs are also laid along the busy town sections" are there any standards or guide lines available in your country for installation of these sign boards!! please take care about the same.

The equation (1) AND 2 the complete notations are missing !! Table 2 what are the units of length(m/km/cm/mm)???

What do you mean by complex network??

Overall not much novelty and in the paper. However the conclusions are very generic and can be re written with respect to the work. the pictures and photographs are not good quality!

It requires a lot of language corrections and improvements.

6. PLOS authors have the option to publish the peer review history of their article (what does this mean?). If published, this will include your full peer review and any attached files.

Reviewer #1: **Yes: **Abdulaziz Alossta

Reviewer #2: No

---

## [Author Response · Author response to Decision Letter 0]

29 Jan 2024

Please see the uploaded file named Response to Reviewers. The comments raised by Reviewers and editors have got replied and implemented, together with the responses to Journal Requirements. Our deepest gratitude goes to you!

---

## [Editor Report · Decision Letter 1]

29 Feb 2024

Strategies to Enhance the Level of Service and Safety of Rural Roads: A Case Study

PONE-D-23-25833R1

Dear Dr. Zhang,

We’re pleased to inform you that your manuscript has been judged scientifically suitable for publication and will be formally accepted for publication once it meets all outstanding technical requirements.

Kind regards,

Ibrahim Badi, PhD

Academic Editor

PLOS ONE
---

## [Editor Report · Acceptance letter]

5 Mar 2024

PONE-D-23-25833R1 

PLOS ONE

Dear Dr. Zhang, 

I'm pleased to inform you that your manuscript has been deemed suitable for publication in PLOS ONE. Congratulations! Your manuscript is now being handed over to our production team.

Kind regards, 

on behalf of

Dr. Ibrahim Badi 

Academic Editor

PLOS ONE